# Effects of Light on Growth and Metabolism of *Rhodococcus erythropolis*

**DOI:** 10.3390/microorganisms10081680

**Published:** 2022-08-20

**Authors:** Selina Engelhart-Straub, Philipp Cavelius, Fabian Hölzl, Martina Haack, Dania Awad, Thomas Brueck, Norbert Mehlmer

**Affiliations:** Werner Siemens-Chair of Synthetic Biotechnology, Department of Chemistry, Technical University of Munich (TUM), 85748 Garching, Germany

**Keywords:** *Rhodococcus*, photo-oxidative stress, stress response, antimicrobial blue light, carotenoids, fatty acids, time-resolved proteomics

## Abstract

*Rhodococcus erythropolis* is resilient to various stressors. However, the response of *R. erythropolis* towards light has not been evaluated. In this study, *R. erythropolis* was exposed to different wavelengths of light. Compared to non-illuminated controls, carotenoid levels were significantly increased in white (standard warm white), green (510 nm) and blue light (470 nm) illuminated cultures. Notably, blue light (455, 425 nm) exhibited anti-microbial effects. Interestingly, cellular lipid composition shifted under light stress, increasing odd chain fatty acids (C15:0, C17:1) cultured under white (standard warm white) and green (510 nm) light. When exposed to blue light (470, 455, 425 nm), fatty acid profiles shifted to more saturated fatty acids (C16:1 to C16:0). Time-resolved proteomics analysis revealed several oxidative stress-related proteins to be upregulated under light illumination.

## 1. Introduction

The genus *Rhodococcus* comprises a cluster of aerobic, non-sporulating, Gram-positive and non-motile bacteria [1,2]. They are ideal candidates for efficient production of a wide range of compounds such as biosurfactants, carotenoids, triacylglycerols or antimicrobials [3]. Furthermore, the genus harbors great potential for finding novel bioactive natural products [4]. The oleaginous *Rhodococcus erythropolis* is known for its high stress tolerance, the ability to metabolize various substrates and the synthesis of carotenoids, resulting in its characteristic coloration [5]. However, little is known about carotenoid function in *R. erythropolis*. 

All aerobic organisms, including *R. erythropolis,* cope with oxidative stress, more specifically, the formation of reactive oxygen species (ROS), including singlet oxygen, superoxide radicals, hydroxyl radicals and hydrogen peroxide. Exposure to visible light and UV irradiation results in higher levels of oxidative stress (photo-oxidative stress). One mechanism of increased ROS formation under light is mediated by intracellular photosensitizers, which absorb incoming light-energy at certain wavelengths and by transfer of excitation energy onto molecular oxygen to generate singlet oxygen [6,7,8]. Some photosensitizers also react by transferring electrons onto oxygen, forming superoxide radicals, hydrogen peroxide and hydroxyl radicals. These, in turn, may trigger damage of cellular components, such as proteins and lipids [9]. Notably, not only UV-light but also blue light exhibits anti-microbial effects against several bacterial species [7,10]. Especially, the wavelengths of 405 nm [11,12], 425 nm [13] and 470 nm [11,12] have been evaluated in previous studies. Here, porphyrins and riboflavins are hypothesized to act as endogenous photosensitizers for blue light [7,10,11,12,13,14].

Cellular detoxification from ROS is mainly performed by oxidoreductases, specifically catalases and peroxidases, which are able to degrade reactive peroxides. Furthermore, highly reactive radicals can be directly deactivated by molecular acceptors, which include different groups of non-protein compounds [6,7,8]. Among these groups, carotenoids can be found in almost all photosynthetic-, as well as various species of non-photosynthetic organisms such as *R. erythropolis*. So far, most studies focused on their function in light harvesting, where they help to efficiently utilize incoming light via absorption of the blue-green light spectrum followed by singlet-singlet energy transfer onto chlorophyll. They are also known for their antioxidant properties. Carotenoids act as quenchers for certain photosensitizers (e.g., triplet chlorophyll) and can directly inactivate different ROS [6]. Similarly, in non-photosynthetic carotenoid-producing organisms, carotenoids serve as protectors against photo-oxidative stress. To that end, photodynamic damaging effects are reduced due to their ability to absorb green and blue light, to react with light-excited photosensitizers, thus preventing increased ROS formation, and their ability to minimize singlet oxygen and radical damage [6,8,15,16].

Enhanced stress resistance can also be conferred through modification of membrane properties, by shifting the ratio between saturated, unsaturated and branched fatty acids (FAs) [17,18,19]. *Rhodococcus opacus*, a close relative of *R. erythropolis*, accumulates high amounts of polyunsaturated fatty acids in response to various stressors, such as salt or ROS [20,21,22]. When *R. erythropolis* was subjected to non-optimal conditions of pH and temperatures, increased FA saturation could be detected [17]. A previously reported stress-related shift in lipid composition is catalyzed by the cyclopropane-fatty acid-acyl phospholipid synthase in *E. coli* and other bacteria, which enables direct modification of membrane lipids by adding cyclopropane moieties to unsaturated FA bonds. These changes are known to influence the physiochemical properties of the membrane (e.g., permeability and fluidity), affecting the cell’s resistance to different stress conditions, including osmotic and oxidative stress, high pressure and temperature change [18,19].

In this study, the photo-oxidative stress response of *R. erythropolis* was investigated by exposure to a selected set of wavelengths of light under aerobic growth conditions. In order to unravel the extensive light stress adaptations, alterations in growth characteristics, carotenoid accumulation and FA profile were monitored. Additionally, quantitative time-resolved proteomics enabled identification of differential levels of proteins and altered regulation of metabolic pathways under light-induced conditions. 

## 2. Materials and Methods

### 2.1. Bacterial Strain and Culture Conditions

*R. erythropolis* JCM3201 (DSM No. 43066, German Collection of Microorganisms and Cell Cultures GmbH) was maintained on Luria-Bertani (LB) agar plates (10 g L^−1^ peptone, 5 g L^−1^ yeast extract, 10 g L^−1^ sodium chloride, 14 g L^−1^ agar). For seed cultures, single colonies were initially cultured in 500 mL baffled shaking flasks holding 100 mL LB liquid medium (10 g L^−1^ peptone, 5 g L^−1^ yeast extract, 10 g L^−1^ sodium chloride) in a rotary incubator (New Brunswick InnovaTM 44, Eppendorf, Hamburg, Germany) for 48 h. All cultivations were performed in 500 mL baffled shaking flasks holding 100 mL LB. Biological triplicates were inoculated to OD_600nm_ 0.5 and cultivated at 28 °C and 120 rpm.

### 2.2. Cultivation of R. erythropolis under Light of Different Wavelenths

To determine growth as well as lipid and carotenoid accumulation, experiments were conducted with different wavelengths of light (LED): blue (425, 455 and 470 nm), green (510 nm), red (680 nm) as well as standard warm white (SWW) light. Control samples were collected from cultures incubated in the dark. Each setting was adjusted to an equal energy output of 236 W m^−2^. Light spectra for each experimental setup are depicted in Appendix A (Appendix A). The light cultivation setup was a customized shaker unit, with individual, bottom-up LED illumination of flasks as described by Paper et al. [23]. Cross-illumination was prevented by shading individual flasks with black paper (Appendix A). The cultivation setup for control samples was a separate shaker (New Brunswick InnovaTM 44, Eppendorf, Hamburg, Germany) lacking the light system. To ensure light-free conditions, the flasks were also shaded with black paper, and the glass door of the shaker was covered with aluminum foil. Experiments were performed in 500 mL baffled shaking flasks with Duran GL32 Membrane Vented Screw caps (DWK Life Science, Wertheim, Germany). Sampling for OD_600nm_ was performed twice a day. Samples for dry cell weight (DCW), carotenoid titers and FA profile were collected after 40 h and once after 122 h and 94 h.

### 2.3. Cultivation of R. erythropolis under LED Light for Proteomic Analysis

To determine growth and carotenoid titers and to conduct proteomics analysis, cultures of *R. erythropolis* were exposed to white LEDs. Setup consisted of a LED Mini- Matrices (Spectral color of 6500 K, max. 750 μmol m^–2^ s^–1^, 504 LEDs, 27 × 42 cm, 24V; LUMITRONIX^®^ LED-Technik GmbH, Hechingen, Germany) installed in a rotary shaker (New Brunswick InnovaTM 42, Eppendorf, Hamburg, Germany) at a height of 30 cm over the incubator platform. Control samples were collected from cultures incubated in the dark and samples from all cultures were collected after 40 h and 122 h.

### 2.4. Growth Analysis

Optical density was measured at 600 nm in a photometer (Nano Photometer NP80, IMPLEN, Munich, Germany) in standard semi-micro cuvettes made of polystyrene holding sample volumes of 1 mL. 

DCW analysis, 25 mL culture was collected. Subsequently, cultures were centrifuged (3500× *g*, 10 min), and cells were washed and lyophilized (−80 °C, ≥ 72 h). Gravimetric measurements were carried out, and the weight of empty vessels was subtracted from weight of vessels containing lyophilized samples.

A light microscope (Motic BA310E) equipped with a Moticam 5.0 MP (Moticeurope, Barcelona, Spain) was used to evaluate cell morphology and contamination.

### 2.5. Pigment Extraction

For pigment extraction from dry biomass, 15 mg of lyophilized biomass was transferred into a reaction tube and mixed with glass beads (0.5 mm) and 1 mL of HPLC-grade acetone. Samples were vortexed horizontally for 10 min and centrifuged at 8000× *g* for 5 min. A volume of 700 µL of the supernatant was carefully transferred into a glass tube and tightly closed. Reaction tubes and glass tubes were wrapped in aluminum foil to avoid light exposure. Carotenoid levels were measured at an absorbance of 454 nm (Nano Photometer NP80, IMPLEN, Munich, Germany), using a solvent-stable cuvette. 

For pigment extraction from wet biomass used in proteomic analysis, a volume of 7.5 mL cell suspension was centrifuged and washed with ddH_2_O. Glass beads (0.5 mm) and 1.5 mL hexane: acetone: ethanol (2:1:1—*v*/*v*/*v*) were added to the cell pellet. The suspension was vortexed horizontally for 30 min to lyse the cells. The liquid phases were separated by centrifugation at 20,000× *g* for 10 min. The hexane phase was collected and carotenoids levels were directly measured in a solvent-stable cuvette in a photometer at 454 nm (UV/Vis spectrophotometer Hewlett Packard 8453, HP, Palo Alto, CA, USA).

### 2.6. Fatty Acid Profile

Lipids extracted from lyophilized biomass were converted into fatty acid methyl ester (FAME) and then analyzed by gas chromatography. In detail, lyophilized biomass (5 mg) was transferred into 10 mL glass vials, crimped with bimetallic lid including a septum (Macherey-Nagel, Düren, Germany). The MultiPurposeSampler MPS robotic (Gerstel, Linthicum Heights, MD, USA), equipped with QuickMix, CF200, Agitator/Stirrer was used for methyl esterification of the intracellular triacylglycerides (TAGs). For quantification, an internal standard of 10 g L^−1^ glyceryl tridodecanoate (C12:0; Sigma-Aldrich, St. Louis, MO, USA) solved in toluol as stock solution was prepared and a volume of 490 µL toluol and 10 µL internal standard were added to the biomass and mixed for 1 min at 1000 rpm. Then, 1 mL of 0.5 M sodium methoxide in methanol was added and the vial was vortexed at 750 rpm and 80 °C for 20 min. The solution was cooled at 5 °C for 5 min, then 1 mL of 5% HCl in methanol (Supleco 17935 solution, Merck AG, Darmstadt, Germany) was added, and the mixture was vortexed at 750 rpm and 80 °C for 20 min. Subsequently, the mixture was cooled at 5 °C for 5 min. A volume of 400 µL ddH_2_O was added and the mixture was vortexed at 1000 rpm for 30 s, then 1 mL hexane was added. FAMEs were extracted by three equal intervals of intermittent shaking for 12 s at 2000 rpm. Then, the vial was centrifuged at 1000 rpm for 3 min. After the vial was cooled at 5 °C for 1 min, 200 µL of the organic phase was transferred to micro vials (Macherey-Nagel, Düren, Germany). 

A GC-2025 coupled to AOC-20i Auto injector and AOC-20s Auto sampler (Shimadzu, Duisburg, Germany) and a flame ionization detector (FID) was used for the ana-lysis and quantification of the FAMEs [24,25]. The injection temperature was 240 °C with a split ratio of 10 and a purge flow of 3 mL min^−1^ helium. Injection volume for all samples was 1 µL. A Zebron ZB-wax column (Phenomenex, Aschaffenburg, Germany) (30 m × 0.32 mm, film thickness 0.25 μm) was used for separation with an initial oven temperature of 150 °C. This temperature was held for 1 min before increasing at a rate of 5 °C min^−1^ up to a final temperature of 240 °C, which was held for 6 min. As carrier gas, hydrogen with a constant flow rate of 3 mL min^−1^ was used. FID was measured at 245 °C with a hydrogen flow of 40 mL min^−1^, synthetic air flow of 400 mL min^−1^ and nitrogen as make-up gas at 30 mL min^−1^. Identification was confirmed with Marine Oil FAME mix (20 components from C14:0 to C24:1; Restek GmbH, Bad Homburg, Germany) and FAME #12 mix (C13:0, C15:0, C17:0, C19:0, C21:0; Restek GmbH, Bad Homburg, Germany) as standards. Normalization was based on the internal methyl laurate (C12; Restek GmbH, Bad Homburg, Germany) standard. Calibration measurements with marine oil mix were performed with 20, 5, 1, 0.5, 0.1 mg mL^−1^. For calibration with FAME #12 5, 2.5, 1.25, 0.5, 0.25, 0.05, 0.01 mg mL^−1^ were used, with methyl laurate 2, 1, 0.2, 0.05, 0.01, 0.002 mg mL^−1^ were used. This allowed comparative quantitation. FA profiles were calculated as percent of total FA content (*w*/*w*).

### 2.7. Proteomics

#### 2.7.1. Protein Extraction and Precipitation

After cell harvest, samples were strictly kept on ice or at 4 °C in the centrifuge. Bacterial cells from 25 mL cultures were pelleted and washed twice with ddH_2_O and centrifugation at 8000× *g* for 10 min. Lysis and extraction were aided by Protein Extraction Reagent Type 4 (Sigma-Aldrich, St. Louis, MO, USA) (1:3, *v*/*v*) and glass beads. Samples were vigorously vortexed for 30 min, then incubated in an ultrasonic bath for 60 min (Ultrasonic Cleaner UCD—THD, VWR, Radnor US). After centrifugation at 13,750× *g* for 30 min, protein precipitation was achieved by mixing the supernatant 1:1 (*v*/*v*) with 20% trichloric acid (*v*/*v*) in HPLC-grade acetone (*v*/*v*) supplemented with 10 mM dithiothreitol (DTT). The mixture was then vortexed and incubated for 1 h, at −20 °C. After centrifugation at 13,750× *g* for 10 min, the protein pellet was washed twice with 1 mL of acetone supplemented with 10 mM DTT. The pellet was air-dried and then dissolved in 8 M urea solution supplemented with 10 mM DTT. Three biological and two technical replicates were prepared for every condition.

#### 2.7.2. Protein Quantification and SDS-PAGE

Protein concentration was determined using a Nanophotometer (NanoPhotometer NP80, Implen GmbH, München, Germany) at 280 nm absorbance. To visually assess the qualitative variances in protein levels, protein extracts were separated on a 12% one-dimensional SDS polyacrylamide gel, according to Awad et al. [26].

#### 2.7.3. In-Gel Digestion of Protein Samples and LC-MS/MS Analysis

In-gel digestion of protein samples and LC–MS/MS analysis, using a timsTOF Pro mass spectrometer equipped with a NanoElute LC System (Bruker Daltonik GmbH, Bremen, Germany) on a Aurora column (250 × 0.075 mm, 1.6 μm; IonOpticks, Hanover St., Australia), was carried out according to the method of Fuchs et al. [27] with the following modifications: Short 12% SDS polyacrylamide gel was used instead of 10% Criterion™ Tris–HCl Protein Gel. The mobile phase comprised two mixtures for reverse-phase separation: 0.1% (*v*/*v*) formic acid—2% (*v*/*v*) acetonitrile—water mixture (A) and a 0.1% (*v*/*v*) formic acid—acetonitrile mixture (B), which was added as a binary gradient at a flow rate of 0.4 µL min^−1^. A separation cycle of 120 min (linearly: 2–17% B in 60 min, 17–25% B in 30 min, 25–37% B in 10 min, 37–95% B in 10 min, maintaining B at 95% for 10 min) was performed. To allow measurement normalization, three QC samples, prepared by mixing 1 µL of each sample, were analyzed at equal intervals between samples (first, mid and last). 

#### 2.7.4. Bioinformatics Analysis

PEAKS Studio software 10.6 (Bioinformatics Solutions Inc., Waterloo, ON, Canada) [28,29,30] was used for peptide and subsequent protein identification. *R. erythropolis* JCM3201 protein (fasta) database based on genome assembly was obtained from NCBI (https://www.ncbi.nlm.nih.gov/genome/?term=txid1833[orgn], accessed on 10 May 2022, 5954 proteins). Search parameters included a precursor mass of 25 ppm using monoisotopic mass and fragment ion of 0.05 Da. Trypsin was selected as digestion enzyme, and a maximum two missed cleavages per peptide were allowed. FDR was set to 1.0%, and search was limited to at least 1 unique peptide per identified protein. The different groups were compared using the Quantification tool PEAKSQ, with a mass error tolerance of 20.0 ppm, Ion Mobility Tolerance of 0.05 Da and a Retention Time Shift Tolerance of 6 min (Auto Detect). Fold change and significance were set to 2 and all proteins were exported.

KOALA (KEGG Orthology And Links Annotation, https://www.kegg.jp/blastkoala/, accessed on 4 July 2022) was used for functional characterization of exported protein sequences [31]. Additionally, annotations were further manually validated with NCBI and Uniprot databases.

## 3. Results

### 3.1. Influence of Light Quality on Growth Characteristics and Carotenoid Accumulation

Throughout the cultivation period, distinct differential growth patterns amongst the various samples were apparent. Controls grown in dark condition as well as red-light-illuminated (680 nm) samples peaked at 24 h with an OD_600nm_ of 9. In contrast, cultures grown under SWW and green light (510 nm) illumination and to a higher extent blue light (455 nm) illumination exhibited notable reduction in growth compared to the former two sample groups (Figure 1a). The same effect could be observed during early stationary phase (40 h). Controls and red-light-illuminated samples reached an OD_600nm_ of 8.4 and 8.3, respectively, while cultures illuminated with different wavelengths stagnated at an OD_600nm_ of 5.7 and 6.3 for white and green light, respectively and an OD_600nm_ of 4.5 for blue light. At late stationary phase (122 h), all samples exhibited a slight but proportional decline in growth, with OD_600nm_ of 6.3 for controls and red-light-illuminated samples, 5.3 and 4.8 for cultures grown under white and green light, respectively, and 4.1 under blue light illumination. 

DCW was determined after 40 and 122 h cultivation. No significant change in biomass formation was observed between control, white- and red-light-illuminated samples, with the latter reaching the highest DCW amongst all three samples at both time points (1.8 and 2.1 g L^−1^, respectively). Samples grown under green light conditions exhibited a slightly lower biomass formation (1.4 and 1.5 g L^−1^, at 40 and 122 h, respectively). Blue-light-illuminated samples formed the lowest DCW of 1.0 g L^−1^ following 122 h of cultivation (Figure 1c).

Carotenoid levels also varied between 40 and 122 h. Cultures illuminated with green light exhibited a significant increase in carotenoid levels after 40 h (0.0145 Abs_454nm_mg^−1^_DCW_) compared to control samples grown under dark condition and samples illuminated with red light (0.0097 Abs_454nm_mg^−1^_DCW_ and 0.0093 Abs_454nm_mg^−1^_DCW_, respectively). Blue light illuminated samples deviated slightly from controls after 40 h of cultivation (0.0081 Abs_454nm_mg^−1^_DCW_). After 122 h, carotenoid level of samples under blue light illumination (0.0054 Abs_454nm_mg^−1^_DCW_) were significantly lower when compared to the control (0.0127 Abs_454nm_mg^−1^_DCW_). While carotenoid levels in samples illuminated with blue light decreased, an increase was observed in the control. This increase was also detected in red-light-illuminated samples, reaching comparable carotenoid levels to samples illuminated with white light. Samples under white light illumination exhibited only minor deviation from 40 h to 122 h. Notably, samples grown under green light had decreased carotenoid levels, reaching comparable values as dark, red and white light illuminated samples.

To further investigate the effect of blue light on growth and pigment formation in *R. erythropolis*, two additional wavelengths of blue light (425 and 470 nm) were evaluated. Since carotenoids reached similar levels in the late stationary phase (122 h), samples were collected at 94 h, to narrow the time frame during which these changes take place. Cells cultivated under 425 nm illumination exhibited severely reduced growth (OD_600nm_ 3.4 and 3.2, after 40 and 94 h, respectively) when compared to controls (OD_600nm_ 5.9 and 5.0 after 40 h and 94 h, respectively), and also exhibiting more growth impediment than samples under green light conditions (OD_600nm_ 4.1 and 3.5 after 40 h and 94 h, respectively). Illumination with 470 nm wavelength light led to intermediate effects on growth when compared to green and the 455 nm light, and allowed the formation of biomass at levels comparable to non-illuminated samples at 40 h. Interestingly, carotenoid accumulation after 40 h was highly increased in the 470 nm illuminated cultures (0.0160 Abs_454nm_mg^−1^_DCW_), while biomass formation was slightly reduced. In comparison, green-light-illuminated samples exhibited slightly lower carotenoid level (0.0130 Abs_454nm_mg^−1^_DCW_), while the control reached carotenoid level of merely 0.0085 Abs_454nm_mg^−1^_DCW_ (Figure 1b). When compared to the non-illuminated controls, illumination with 425 nm light did not exhibit differential carotenoid level in early stationary phase. After 94 h of cultivation, a significant decrease in carotenoid level was observed (0.0045 Abs_454nm_mg^−1^_DCW_). Interestingly, carotenoid levels of samples illuminated with green light or 470 nm light dropped to comparable levels as white light after 94 h (Figure 1d).

Microscopy of all the *Rhodococcus* cultures showed agglomeration of cells under blue light conditions. While agglomeration was enhanced at 455 nm light illumination, only dense cellular aggregates could be detected in samples illuminated at 425 nm (Figure 2).

### 3.2. Influence of Light Quality on Fatty Acid Composition

The influence of light on the FA profiles were assessed by FAME analysis. The FA content of *R. erythropolis* ranged between 45.6 and 59.4 µg mg^−1^_DCW_ for all cultures. Relative quantification of FAs are depicted as percentage of total FAs (*w*/*w*). Vaccenic acid (C18:1), palmitic acid (C16:0) and palmitoleic acid (C16:1) remained the major FA components. Although the FA profile shifts induced by light stress only slightly diverged over time, the profile shift is most apparent in the increase in odd chain fatty acids (OCFAs; Figure 3). This shift in distribution can be observed as an increase in pentadecanoic acid (C15:0) as well as heptadecenoic acid (C17:1) in cells stressed under white and green light compared to non-illuminated control. After 40 h, OCFAs in cells treated with white light increased to 7.1% (C15:0) and 5.4% (C17:1). OCFAs in cells treated with green light increased to 8.1% (C15:0) and 5.6% (C17:1) compared to the control (2.7% C15:0 and 3.7% C17:1) (Figure 3a). These observations were recorded at all time points; over time, the concentration of OCFAs in the cultures remained nearly constant (Figure 3). Red light showed no significant effect on the FA profile compared to the control. For blue light wavelengths, a shift from C16:1 to C16:0 was detected. Illumination at 425, 455 and 470 nm wavelength resulted in C16:0 content of 24.8%, 25.5% and 24.2% as well as C16:1 content of 12.4%, 12.2% and 12.9%, respectively. In comparison, the control contained 17% of C16:0 and 16.6% of C16:1 at 40 h (Figure 3c). This measurement was consistent with samples collected at 94 h (Figure 3d). 

### 3.3. Effect of Light on Protein Levels

To elucidate the response of *R. erythropolis* to light stress, a time-resolved proteomics approach was implemented. As we were interested in the physiologically relevant stress response to natural light conditions, a daylight LED (6500 K) was chosen for further testing. Carotenoid titers are shown in Appendix A (Appendix A). A total of 3565 proteins of 5954 annotated proteins in the database were identified with at least one unique peptide, corresponding to a coverage of 60%. Time-resolved proteomics revealed 355 proteins (significance ≥ 2, fold change ≥ 2, detected in at least one sample per group, based on LFQ by PEAKS Studio; see Appendix A) as differentially regulated after 40 h under light stress. This set of proteins comprises 141 upregulated and 214 downregulated proteins compared to control (Figure 4a). After 122 h, 704 proteins were differentially regulated, of which 287 proteins were upregulated and 417 proteins downregulated compared to control (Figure 4b). In contrast to the phenotypic observations of increased carotenoid accumulation at 40 h, *R. erythropolis* exhibited a strong response on the proteomic level at 122 h; twice the number of proteins were differentially regulated by light stress, in comparison to 40 h. 

Differentially regulated proteins (fold change and significance ≥ 2) were annotated to molecular functions by KEGG; orthologs may be assigned with the same KEGG Object (KO) identifier. At 40 h, 53.3% of down- and 44.7% of upregulated proteins were matched to KO identifiers. At 122 h, 53.5% of down- and 50.5% of upregulated protein were matched. Fold changes above 1 represent upregulation, while downregulation is illustrated by fold changes below 1.

Table 1 highlights proteins related to the stress response of *R. erythropolis* under light conditions with different abundance compared to the control at 40 and 122 h. Distinctly, four different transcription factors involved in oxidative stress response were identified: WP_020906739.1 (40 h 2.38-fold), WP_019747469.1 (40 h 2.38-fold; 122 h 3.88-fold), WP_060939090.1 (122 h 3.29-fold) and WP_020906601.1 (122 h 4.51-fold). 

Additionally, several enzymes involved in inactivation of ROS were identified: WP_003940303.1 (40 h 3.06-fold; 122 h 2.80-fold), WP_021346030.1 (40 h 2.60-fold) and WP_003942119.1 (40 h 2.35-fold; 122 h 2.16-fold). However, a bifunctional catalase/peroxidase was identified in a cluster of downregulated genes (WP_060938296.1; 40 h 0.45-fold). A second protein, that exhibited a decreased level is WP_019749140.1 (122 h 0.49-fold). Two more stress-related proteins were upregulated: WP_019747464.1 (122 h 2.0-fold), WP_019749386.1 (40 h 2.39-fold). The regulation of WP_003942530.1 inverted during cultivation, displaying downregulation after 40 h and upregulation after 122 h (40 h 0.43-fold, 122 h 2.86-fold).

The shift in FA profile of *R. erythropolis* under light stress towards OCFAs could be connected to differentially regulated proteins involved in the propanoate pathway. At 40 h, 2-oxoisovalerate dehydrogenase E1 component alpha subunit (WP_060938994.1, K00166) and beta subunit (WP_060938993.1, K00167) as well as 2-oxoisovalerate dehydrogenase E2 component (WP_020970089.1, K09699) synthesizing propanoyl-CoA from 2-oxobutanoate were upregulated 3.82-, 2.63- and 3.05-fold, respectively. Dihydrolipoamide dehydrogenase (WP_174531767.1, K00382), which facilitates the reaction of dihydrolipoamide-E to lipoamide-E was downregulated by 0.41-fold. This enzymatic step branches off the propanoyl-CoA production pathway from 2-oxobutanoate. Furthermore, the acetyl-CoA synthetase (WP_019745948.1, K01895) facilitating the reaction from propanoyl-CoA to propanoate is downregulated by 0.46-fold (Figure 5).

At 122 h, the following enzymes remained upregulated (K00166, K00167, K09699): WP_060938994.1, 2.73-fold; WP_060938993.1, 2.62-fold; WP_020970089.1, 3.10-fold. In contrast to this, proteins matched to the same activity and therefore the same KO identifiers were downregulated: WP_060938768.1, WP_060938769.1 and WP_060938770.1 with a 0.42-, 0.23-, 0.35-fold change, respectively. The enzyme bccA (WP_060939079.1) synthesizing malonyl-CoA from acetyl-CoA was upregulated at both time points (40 h 3.28-fold; 122 h 3.92-fold).

Additionally, two enzymes of the cyclopropane-fatty-acyl-phospholipid synthase family were significantly upregulated. After 40 and 122 h, levels of WP_060938639.1 increased 22.02- and 18.92-fold. The level of WP_060938640.1 increased 43.79- and 31.54-fold, respectively (Table 2).

## 4. Discussion

### 4.1. Effects of Light Illumination on Growth

In accordance with previous studies [7,10,11,12,13,14], blue light exhibits antimicrobial activity on *R. erythropolis*. Spectrophotometric (OD_600nm_) data and DCW consistently revealed a significant decrease in bacterial growth when illuminated with 425 or 455 nm blue light. To a lesser extent, cells illuminated by 470 nm light also exhibited reduced growth, with OD_600nm_ values between 455 nm and green light illumination. Wang et al. [11] reported similar results when comparing the effects of 405 nm to 470 nm light illumination on the human pathogen *Neisseria gonorrhoeae*, the causative agent of Gonorrhoeae. Furthermore, Mathews and Sistrom [32] reported that carotenoid-deficient mutants of the pigment-producing non-phototrophic organism *Sarcina lutea* were highly susceptible to sunlight. However, lethal effects could only be observed under aerobic conditions, while wild-type cells were not affected by either condition (light anaerobic condition, light aerobic condition). They concluded that photo-oxidative conditions were required to exhibit antimicrobial effects on *S. lutea,* as stand-alone light stress was not sufficient to cause lethal effects. Furthermore, they stated, that carotenoids can protect cells against photo-oxidative damage, as wild-type was not susceptible to sunlight treatment [32]. This cumulative data suggests that the antimicrobial effect of blue light could, in part, be propagated by endogenous photosensitizer present in *R. erythropolis*. These photosensitizers would absorb light-energy and transfer the excitation energy or electrons onto oxygen, resulting in ROS among other reaction products [6,7,8].

White light and green light also had distinct effects on cell growth. Here, cultures exhibited slightly decreased growth compared to the control for around 24 h from inoculation. After that, differences in growth slowly decreased. This is likely correlated to the induction of intracellular adaptation mechanisms such as increasing carotenoid content, which, following initial adaptation period, enabled growth more similar to those of control samples. Red light did not have an effect on culture growth and carotenoid accumulation.

### 4.2. Changes in Carotenoid Content

Spectrometric analysis of sample extracts after 40 h growth revealed a drastic increase in carotenoid level, particularly when cells were illuminated with blue (470 nm) and green (510 nm) light and to a lesser extend with white light. This observation suggests that carotenoids in *R. erythropolis* serve as protectors against photo-oxidative damage. As carotenoids themselves generally absorb light in the range of 400 to 550 nm, they can protect endogenous photosensitizers from exciting light of that wavelength. Available models describe the reaction of carotenoids with reactive oxygen species (e.g., singlet oxygen), where the former drain the excitation energy from the singlet oxygen and subsequently dissipate the captured energy via heat formation. They might react similarly with different endogenous excited photosensitizers, preventing the formation of ROS. Other models also predict possible reactions with radicals, generating different mechanisms by which these reactions and the subsequent recovery of reacted carotenoids could occur [6,8,15].

Following 122 h of cultivation, 425 nm blue light led to a significant decrease in carotenoid content, consistent with the severe decrease in biomass formation. Growth under dark conditions and red light illumination led to comparable carotenoid levels to those of cultures grown under white or green light. Here, carotenoid levels under dark conditions and red light illumination increased, while samples under green light conditions exhibited decreased carotenoid accumulation. Nutrient limitation, especially nitrogen limitation, can trigger carotenoid accumulation in *Rhodococcus*, this could explain the increase in carotenoids in control and red-light-illuminated samples [33,34]. As nitrogen limitation often occurs simultaneously with oxidative stress, carotenoid accumulation could present as an adaptation response mechanism [35]. This is in line with the 2.0-fold increase in Dps, a starvation-inducible DNA-binding protein, after 122 h. Dps is mainly regulated by nutrient deprivation and also acts in oxidative stress response. In our work, no differential regulation of Dps was observed at 40 h [36,37,38]. As stated earlier, carotenoid level in cells illuminated with green or blue (470 nm) light significantly increased during the first 40 h of cultivation, but decreased in late stationary phase (94 h and 122 h). This change in accumulation levels could occur due to nutrient deficiency in late stationary phase and the resulting energy demand being too severe to maintain these high amounts of carotenoids. Therefore, expendable amounts of carotenoids were degraded. Comparison between samples taken at 122 h and 94 h revealed that the observed decrease in carotenoids in light treated samples already occurred before 122 h. However, an increase in carotenoid content in non-illuminated controls and red-light-illuminated samples did not occur after 94 h of cultivation.

In summary, the detected increase in carotenoids under white, green and blue (470 nm) light represents an early response (40 h) to light stress, while the later increase in carotenoid levels for control and red illuminated samples (122 h) depicts an adaptive response to nitrogen limitation, which occurs at a later point. In accordance to these findings, Cohen et al. [39] observed an increase in carotenoid content in *Rhodococcus* sp. APG1 isolates when grown under light.

### 4.3. Transcription Factors

Proteomics analysis allowed identification of several proteins involved in photo-oxidative stress response. Sigma factor sigF (WP_020906739.1) is part of the heat stress and oxidative stress response in *Mycobacterium smegmatis* [40,41]. Within this studies’ data, sigF displays a 2.38-fold increase in protein level after 40 h; however, it was not differentially regulated after 122 h of cultivation. In contrast, sigB (WP_019747469.1) a light- and heat-stress-related sigma factor, was significantly up-regulated at 40 h and 122 h (2.38-fold and 3.88-fold, respectively). sigB is well characterized in *Listeria monocytogenes* and *Synechocystis* sp. PCC 6803 and is induced as an essential factor during heat-, oxidative- and light-stress response, especially under blue light [42,43,44,45,46]. Furthermore, Hakkila et al. [45] and Turunen et al. [44] reported similar results in the cyanobacterium *Synechocystis* sp. *PCC 6803*. The former group was able to engineer mutants of *S.* sp. *PCC 6803* with deletion mutations in all group 2 σ factors except sigB. The mutants exhibited high carotenoid accumulation and good growth under singlet oxygen and high light stress [45].

Two more proteins belonging to the hspR MerR transcriptional regulator family (WP_060939090.1 and WP_020906601.1) were upregulated after 122 h (3.29- and 4.51-fold) and were recently reported to be involved in hydrogen peroxide-induced oxidative stress response. Additionally, Lu et al. [47] established, that mutants missing hspR activity exhibited greater sensitivity to hydrogen peroxide. Furthermore, they identified *ahpC*, coding for a lipoyl-dependent peroxiredoxin, to be upregulated by hspR activity.

### 4.4. Catalases and Peroxidases

Three enzymes involved in ROS-inactivation were identified as additional indicators of photo-oxidative stress after 40 h. A superoxide dismutase (SOD) (WP_003940303.1) was 3.06-fold upregulated, while a monofunctional catalase (WP_021346030.1; katE) was upregulated 2.60-fold under light compared to the control. SOD converts superoxide radicals to hydrogen peroxide, the catalase reduces two hydrogen peroxide molecules to water and molecular oxygen. *katE* is of major interest, as it is reported to be induced by hydrogen peroxide. On the contrary *katG* (WP_060938296.1), a gen coding for a catalase/peroxidase found in a cluster of downregulated proteins after 40 h (0.45-fold), is reported not to be affected by hydrogen peroxide level [20,35,48].

The third enzyme, ahpC (WP_003942119.1), more specifically a subunit of the respective enzyme exhibited a 2.35-fold increase in protein abundance. This enzyme belongs to a group of peroxidases that utilize cysteine residues for reduction in alkyl-peroxides.

After 122 h, SOD and ahpC levels remained upregulated under light conditions (2.80- and 2.16-fold, respectively); however, no other differentially upregulated catalases were confirmed. Interestingly, another peroxidase (thioredoxin-dependent thiol peroxidase; WP_019749140.1) exhibited a 2-fold decrease in protein level after 122 h. 

### 4.5. Other Proteins Related to Stress

Besides proteins specific to certain types of stress response dnaJ a molecular chaperon is part of the overall stress response [20,35]. However, upregulation was only detected after 40 h (WP_019749386.1 2.39-fold). Furthermore, levels of thioredoxin trxA, another part of an antioxidant system [20,35], was downregulated after 40 h of cultivation (0.43-fold), but exhibited elevated levels after 122 h (WP_003942530.1 2.86-fold). 

Interestingly, a starvation-inducible DNA-binding protein (Dps) (WP_019747464.1) did not differ in protein abundance after 40 h, but increased 2.0-fold after 122 h, indicating severe nutrient deficiency in samples at late stationary phase. Previous reports presented data linking Dps to nutrient deprivation as well as oxidative stress, where its production is induced as part of the stress response [36,37,38].

### 4.6. Influence of Light on Fatty Acid Composition

One aspect of adaptation to various stress conditions takes place at the cell membrane. To maintain biological functions, bacteria adapt their lipid composition to maintain membrane fluidity by changing the level of FA saturation [49,50,51]. Properties, such as membrane fluidity and permeability, are of major importance for cell survival. Temperature fluctuation, oxidation of cellular lipids and other stressors can threaten cell viability and trigger changes in membrane composition, modulating these properties [18,19].

While oxidative stress led to increased lipid accumulation in the closely related *R. opacus* [52], the FA content remained in a similar range of around 5% (g g ^−1^
_DCW_) in all *R. erythropolis* samples subjected to light stress.

After exposure to H_2_O_2_-induced oxidative stress, a decrease in cell size and the tendency to form multicellular conglomerates was observed in *R. opacus* 1CP [22]. In our study, *R. erythropolis* cells subjected to blue light stress also agglomerated (455 nm)/aggregated (425 nm) (Figure 2). In this manner, inner cells of the agglomerate/aggregate could be shielded from damaging blue light [53,54].

The FA composition of *R. erythropolis* shifts towards saturated FAs, especially C16:0, in contrast to the observation reported by Solyanikova et al. [22] for *R. opacus* 1CP, where upon H_2_O_2_ exposure, levels in unsaturated FAs increased. The increase in saturated FA detected in our work is in accordance with Wu et. al., whereby methicillin-resistant *S. aureus* (MRSA) displayed decreased levels of unsaturated FAs, namely, C16:1, C20:1 and C20:4, when illuminated with blue light (415 nm) [55]. It is hypothesized that an increase in saturated FAs in the membrane renders the membrane less permeable [20]. 

The levels of saturated OCFAs, mainly C15:0 and C17:0 (2- to 3-fold), and branched (10-methyl) FAs were reported to be increased in *R. opacus* in the presence of aromatic compounds [56]. In our work, elevated content of OCFAs were detected under green and white light conditions. OCFAs were produced in *Yarrowia lipolytica* and *Rhodococcus* sp. YHY01 when supplemented with propionate as carbon source. Propanoyl-CoA was proposed as primer for OCFA synthesis in *Rhodococcus*, while malonyl-CoA, generated from acetyl-CoA, acts as primer for even chain fatty acid synthesis [57,58]. As a metabolic intermediate, propanoyl-CoA can be synthesized by the degradation of propionate, OCFAs and some amino acids such as cysteine, methionine and threonine. Due to the cellular toxicity of propanoyl-CoA at high concentrations, microorganisms evolved to regulate its accumulation [59,60]. As depicted in Figure 5, 2-oxoisovalerate dehydrogenase E1 component subunits (K00166 and K00167) as well as 2-oxoisovalerate dehydrogenase E2 (K09699), responsible for the synthesis of propanoyl-CoA are upregulated at 40 h and 122 h for cells cultivated under white light. Increased availability of these enzymes could explain the increase in OCFAs, namely, C15:0 and C17:1, in *R. erythropolis* cells. The bidirectional acetyl-CoA synthetase (K01895) could be tightly regulated under feedback mechanisms to limit the adenylation of propanoyl-CoA back to propionate, ensuring ample levels of the building block for OCFA production. The enzyme level is downregulated by 2-fold under light conditions.

Furthermore, time-resolved proteomics revealed a significant increase in cyclopropane-fatty-acyl-phospholipid synthase family proteins. Specifically, levels of two proteins, WP_060938639.1 and WP_060938640.1, exhibited a drastic 22.0-fold and 43.8-fold increase after 40 h under light conditions and an 18.9-fold and 31.5-fold increase after 122 h, respectively. These enzymes are responsible for modification of FA double bonds by adding cyclopropyl moieties, a modification that was found to increase resistance to different stresses in *E. coli* and other bacteria, namely, high pressure, acidity and heat resistance [18,19]. Our work demonstrates that protein production of this group of enzymes is also subject to light stress. The effect of heat or temperature stress, especially in combination with light stress, could be further investigated in *R. erythropolis*.

## 5. Conclusions

The results obtained in this study provide insight into adaptation mechanisms of *R. erythropolis* when subjected to photo-oxidative stress. To that end, an increase in stress-related sigma factors and other proteins, specifically peroxidases, was detected by time-resolved proteomic analysis. Most proteins discussed here were directly linked to oxidative stress specifically, indicating that light stress in *R. erythropolis* likely acts via endogenous photosensitizers and oxygen. Blue light (425, 455 nm) significantly curbed growth, while green (510) and white (SWW) light led to slightly reduced growth compared to control under dark conditions, further adding to previous studies investigating antimicrobial properties blue light. Blue light of 470 nm wavelength exhibited an intermediate effect between white/green and blue (455, 425 nm) light. Additionally, carotenoid levels were found to be increased in samples illuminated with SWW light and to much higher extent when treated with green (510 nm) and blue (470 nm) light. In regard to previous studies, these results imply carotenoids are synthesized as part of a photo-oxidative stress response. Stress adaptations further involved a shift in FA composition towards OCFAs as well as saturated and branched FAs, likely altering membrane properties.

## Figures and Tables

**Figure 1 microorganisms-10-01680-f001:**
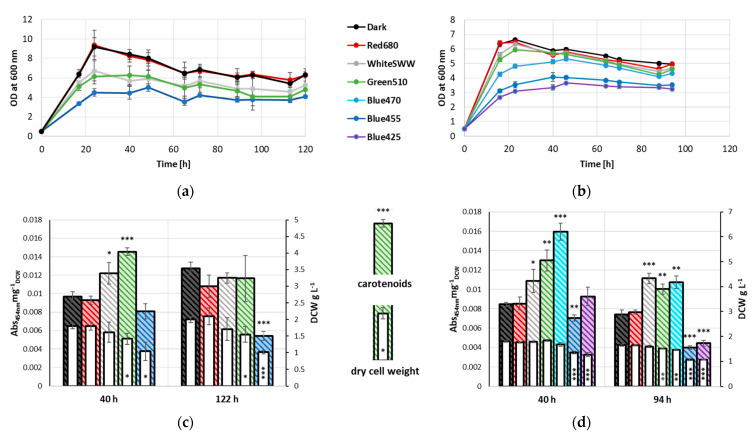
Growth of *R. erythropolis* measured as OD at 600 nm at 236 W s^−2^ illumination with white (standard warm white), blue (455 nm), green (510 nm) and red (680 nm) light as well as control cultivated under exclusion of light, n = 3. (**a**) Growth for 122 h. (**b**) Growth for 94 h with two additional wavelengths (425, 470 nm) of blue light; biomass formation (dry cell weight) and carotenoid accumulation (normalized to dry cell weight) of *R. erythropolis* at 236 W s^−2^ illumination with white (SWW), blue (455 nm), green (510 nm) and red (680 nm) light as well as a control, * *p* < 0.05, ** *p* < 0.01, *** *p* < 0.001 (t-tests evaluated against respective non-illuminated control), n = 3. (**c**) Biomass formation and carotenoid accumulation of *R. erythropolis* at 40 h and 122 h. (**d**) Biomass formation and carotenoid accumulation of *R. erythropolis* at 40 h and 94 h with two additional wavelengths (425, 470 nm) of blue light.

**Figure 2 microorganisms-10-01680-f002:**
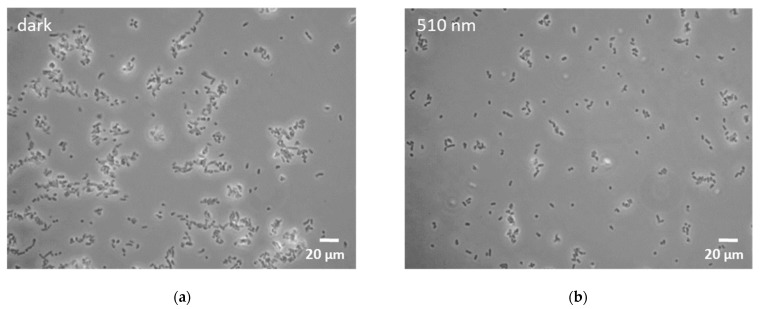
Microscopy images of *R. erythropolis* exposed to selected light conditions for 46 h. (**a**) Non-illuminated control; (**b**) green light (510 nm); (**c**) blue light (455 nm); (**d**) blue light (425 nm). Scale bar indicates a length of 20 µm.

**Figure 3 microorganisms-10-01680-f003:**
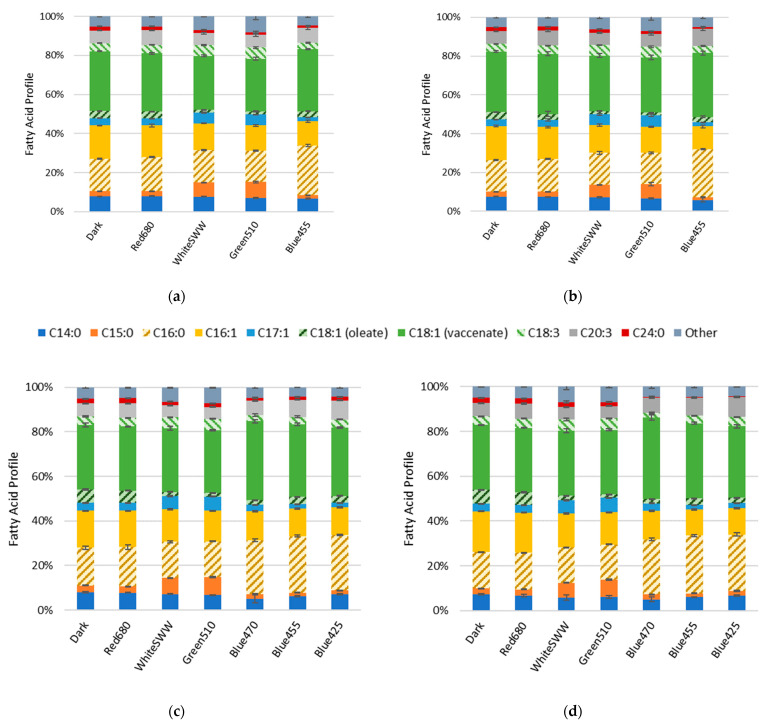
Fatty acid profiles of *R. erythropolis* at 236 W s^−2^ illumination with white (standard warm white), blue (425, 455 and 470 nm), green (510 nm) and red (680 nm) light as well as a control (n = 3). “Other” constitutes fatty acids with a representation below 2% of total fatty acid content (*w*/*w*), and include C14:1, C17:0, C18:0, C20:1, C20:5, C22:1. (**a**) Fatty acid profile collected at 40 h, (**b**) at 122 h, (**c**) at 40 h, (**d**) and at 94 h.

**Figure 4 microorganisms-10-01680-f004:**
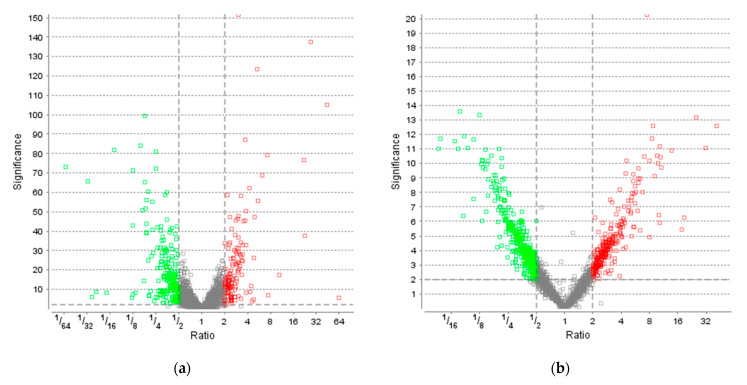
Volcano plots of proteins extracted from cultures grown under LED light compared to the control. Analysis was performed based on label-free quantification (LFQ). The significance is plotted against the fold change ratio. The non-axial vertical lines denote fold change thresholds of 2 (upregulated ≥ 2 in red and downregulated ≤ 0.5 in green), the non-axial horizontal lines denote a significance threshold of 2. Figure compiled by PEAKS Studio Xpro. (**a**) Volcano plot after 40 h. (**b**) Volcano plot after 122 h.

**Figure 5 microorganisms-10-01680-f005:**
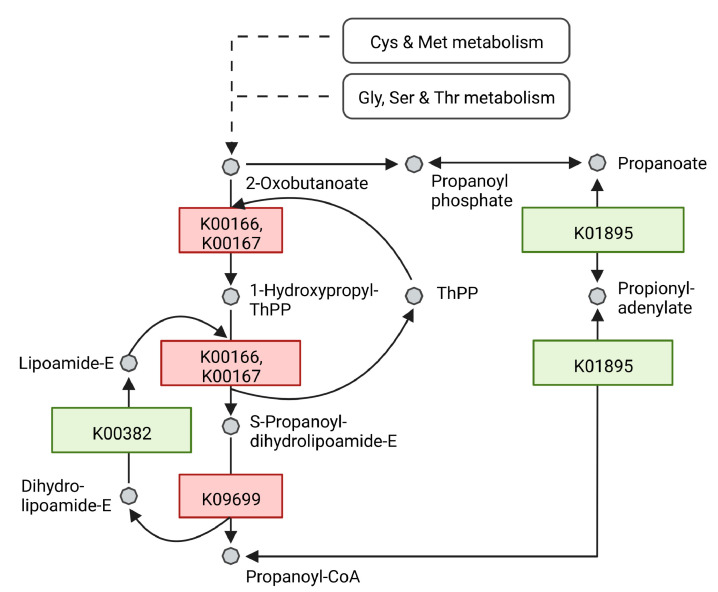
Schematic illustration of an excerpt from the propanoate metabolism in *R. erythropolis*. After 40 h of growth, proteins which are at least 2-fold upregulated by light stimulation are depicted in red, proteins downregulated at least 2-fold are depicted in green. Each enzyme is represented with its respective KO identifier. Cys = Cysteine, Met = Methionine, Gly = Glycine, Ser = Serine, Thr = Threonine. Adapted from KEGG pathway 00640 Propanoate metabolism. Created with BioRender.com.

**Table 1 microorganisms-10-01680-t001:** Selected proteins involved in oxidative stress discussed here. Protein abundance and fold change in samples grown under white light were compared to non-illuminated control. NCBI accession numbers as well as matched KO identifiers are listed. Further identified unique peptides, significance and fold change at 40 h and 122 h of samples are stated. Fractions below 1 depict downregulation. n.d. = not detected.

Transcription Factors Involved in Oxidative Stress								
			40 h			122 h		
Accession Number	Description	KO-ID	Identified Unique Peptides	Significance	Fold Change	Identified Unique Peptides	Significance	Fold Change
WP_019747469.1	SigB; RNA polymerase sigma-B factor	K03090	1	3.8	2.38	1	2.2	3.88
WP_020906601.1	hspR; MerR family transcriptional regulator, heat shock protein hspR	K13640	2	2.36	1.38	1	4.19	4.51
WP_020906739.1	MULTISPECIES: RNA polymerase sigma factor SigF [*Rhodococcus*]	K03090	1	4.35	2.38	n.d.	n.d.	n.d.
WP_060939090.1	hspR; MerR family transcriptional regulator, heat shock protein hspR	K13640	5	4.07	1.39	4	4.52	3.29
**Catalases and Peroxidases**								
WP_019749140.1	BCP, PRXQ, DOT5; thioredoxin-dependent peroxiredoxin [EC:1.11.1.24]	K03564	8	0.11	0.99	8	3.13	0.49
WP_021346030.1	katE, CAT, catB, srpA; catalase [EC:1.11.1.6]	K03781	14	26.31	2.6	14	0.19	1.06
WP_003940303.1	SOD; superoxide dismutase, Fe-Mn family [EC:1.15.1.1]	K04564	5	47.57	3.06	3	4.34	2.8
WP_003942119.1	ahpC; lipoyl-dependent peroxiredoxin subunit C [EC:1.11.1.28]	K24126	16	47.31	2.35	16	3.56	2.16
WP_060938296.1	katG; catalase-peroxidase [EC:1.11.1.21]	K03782	27	23.13	0.45	24	1.78	0.63
**Other Stress-Related Proteins**								
WP_003942530.1	trxA; thioredoxin	K03671	2	9.59	0.43	3	4.05	2.86
WP_019747464.1	dps; starvation-inducible DNA-binding protein	K04047	14	0.32	0.98	14	2.98	2.0
WP_019749386.1	dnaJ; molecular chaperone DnaJ	K03686	12	5.42	2.39	13	0.78	1.28

**Table 2 microorganisms-10-01680-t002:** Selected proteins involved in propanoate metabolism and cyclopropane-fatty-acyl-phospholipid synthesis discussed here. Protein abundance and fold change in samples grown under white light were compared to non-illuminated control. NCBI Accession numbers as well as matched KO identifiers are listed. Further identified unique peptides, significance and fold change at 40 h and 122 h of samples are stated. Fractions below 1 depict downregulation. n.d. = not detected.

Propanoate Pathway								
			40 h			122 h		
Accession Number	Description	KO-ID	Identified Unique Peptides	Significance	Fold Change	Identified Unique Peptides	Significance	Fold Change
WP_019745948.1	acs; acetyl-CoA synthetase [EC:6.2.1.1]	K01895	20	26.37	0.46	15	13.6	0.08
WP_020970089.1	bkdB; 2-oxoisovalerate dehydrogenase E2 component (dihydrolipoyl transacylase) [EC:2.3.1.168]	K09699	7	29.17	3.05	7	4.78	3.1
WP_060938768.1	bkdA1; 2-oxoisovalerate dehydrogenase E1 component alpha subunit [EC:1.2.4.4]	K00166	22	20.92	0.59	19	4.81	0.42
WP_060938769.1	bkdA2; 2-oxoisovalerate dehydrogenase E1 component beta subunit [EC:1.2.4.4]	K00167	11	28.47	0.57	11	8.94	0.23
WP_060938770.1	bkdB; 2-oxoisovalerate dehydrogenase E2 component (dihydrolipoyl transacylase) [EC:2.3.1.168]	K09699	21	23.64	0.55	18	5.08	0.35
WP_060938993.1	bkdA2; 2-oxoisovalerate dehydrogenase E1 component beta subunit [EC:1.2.4.4]	K00167	11	30.2	2.63	12	4.09	2.62
WP_060938994.1	bkdA1; 2-oxoisovalerate dehydrogenase E1 component alpha subunit [EC:1.2.4.4]	K00166	12	50.3	3.82	13	4.41	2.73
WP_060939079.1	bccA, pccA; acetyl-CoA/propionyl-CoA carboxylase, biotin carboxylase, biotin carboxyl carrier protein [EC:6.4.1.2 6.4.1.3 6.3.4.14]	K11263	31	58.35	3.28	34	5.63	3.92
WP_060939587.1	acs; acetyl-CoA synthetase [EC:6.2.1.1]	K01895	23	33.08	2.28	20	4.29	2.39
WP_174531767.1	DLD, lpd, pdhD; dihydrolipoamide dehydrogenase [EC:1.8.1.4]	K00382	22	30.59	0.41	16	8.77	0.19
**Cyclopropane-Fatty-Acyl-Phospholipid Synthase Family**								
WP_060938639.1	cfa; cyclopropane-fatty-acyl-phospholipid synthase [EC:2.1.1.79]	K00574	2	76.43	22.02	1	6.24	18.92
WP_060938640.1	cfa; cyclopropane-fatty-acyl-phospholipid synthase [EC:2.1.1.79]	K00574	3	105.03	43.79	4	11.05	31.54

## Data Availability

Not applicable.

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
