# Peer review of "Effects of Light on Growth and Metabolism of Rhodococcus erythropolis"

_microorganisms, 2022, doi:10.3390/microorganisms10081680_

Round 1

Reviewer 1 Report

This paper is suirable for the publication. The manuscript describes up-to-date information about R. erythropol. Modern methods were used, including proteomic studies. The data obtained contribute to the understanding of adaptive mechanisms of R. erythropol to light of different wavelengths.

Author Response

The authors would like to thank the reviewer for the comments. We have revised the manuscript with the help of a native English speaker. We hope the improved manuscript is suitable for publication in the journal of Microorganisms.

Reviewer 2 Report

The manuscript entitled “Effects of light on growth and metabolism of Rhodococcus  erythropolis” described the novel data about insight into the adaptation mechanisms of R. erythropolis when subjected to photo-oxidative stress.

The study can be accepted after minor revision.

1) The insufficient quality of the captions to the figures should be noted, as well as the lack of size scales, and units of measurement (Fig.2, etc.).

2) Do the data found by the authors on changes in carotenoids agree with the literature data?

3) In the Introduction, the authors mention blue light together with a rather wide range of wavelengths. Could the authors, please, clearly indicate the borders of the blue light range?

Author Response

General comment: The manuscript entitled “Effects of light on growth and metabolism of Rhodococcus erythropolis” described the novel data about insight into the adaptation mechanisms of R. erythropolis when subjected to photo-oxidative stress. The study can be accepted after minor revision.

The authors would like to thank the reviewer for his/her valuable input to improve our manuscript. We hope the improved manuscript is suitable for publication in the journal of Microorganisms. We have implemented the missing information into our manuscript. Additionally, you can find the point-to-point answers below.

Comment 1: The insufficient quality of the captions to the figures should be noted, as well as the lack of size scales, and units of measurement (Fig.2, etc.).

We thank the reviewer for the comment. We have revised the figure captions and enhanced their quality. We further included the missing scale bars.

Comment 2: Do the data found by the authors on changes in carotenoids agree with the literature data?

We thank the reviewer for the comment. The observed changes in carotenoids are indeed in accordance with literature. We now added this information into the discussion part.

Comment 3: In the Introduction, the authors mention blue light together with a rather wide range of wavelengths. Could the authors, please, clearly indicate the borders of the blue light range?

We thank the reviewer for the comment. We revised the mentioned section of the introduction in order to clarify and listed the specific wavelengths tested in literature.